# Development of Organ-Preserving Radiation Therapy in Gastric Marginal Zone Lymphoma

**DOI:** 10.3390/cancers14040873

**Published:** 2022-02-10

**Authors:** Daniel Rolf, Gabriele Reinartz, Stephan Rehn, Christopher Kittel, Hans Theodor Eich

**Affiliations:** Department of Radiation Oncology, University Hospital of Münster, 48149 Münster, Germany; daniel.rolf@ukmuenster.de (D.R.); gabriele.reinartz@ukmuenster.de (G.R.); stephan.rehn@ukmuenster.de (S.R.); Christopher.Kittel@ukmuenster.de (C.K.)

**Keywords:** non-Hodgkin lymphoma, gastric marginal zone lymphoma, MALT lymphoma of the stomach, radiation therapy, ILROG

## Abstract

**Simple Summary:**

Gastric marginal zone lymphoma of the stomach is a rare cancer type primarily treated with oral proton pump inhibitors. If the disease does not respond to this, radiation is the treatment of choice. This review presents the development of radiation therapy over the last decades. Earlier, the stomach was surgically removed and irradiation was performed using large-field techniques and high doses of radiation. Currently, the standard treatment is the use of small-volume radiation therapy (with few side effects) with the preservation of the stomach, which provides excellent outcomes. In addition, this paper provides an outlook on current studies and possible future developments.

**Abstract:**

Gastric marginal zone lymphoma (gMZL) of mucosa-associated lymphoid tissue (MALT) may persist even after *H. pylori* eradication, or it can be primarily *Helicobacter pylori* (*H. pylori*) independent. For patients without the successful eradication of lymphoma, or with progressive disease, treatment options have historically included partial or total gastrectomy. Presently, in these instances, curative radiation therapy (RT) is the current standard of care. This review emphasizes the historically changing role of radiation therapy in gMZL, progressing from large-volume RT without surgery, to localized RT, on its own, as a curative organ-preserving treatment. This overview shows the substantial progress in radiation therapy during the recent two to three decades, from high-dose, large-field techniques to low-dose, localized target volumes based on advanced imaging, three-dimensional treatment planning, and advanced treatment delivery techniques. RT has evolved from very large extended field techniques (EF) with prophylactic treatment of the whole abdomen and the supradiaphragmatic lymph nodes, applying doses between 30 and 50 Gy, to involved-field RT (IF), to the current internationally recommended involved site radiation therapy (ISRT) with a radiation dose of 24–30 Gy in gMZL. Stage-adapted RT is a highly effective and safe treatment with excellent overall survival rates and very rare acute or late treatment-related toxicities, as shown not only in retrospective studies, but also in large prospective multicenter studies, such as those conducted by the German Study Group on Gastrointestinal Lymphoma (DSGL). Further de-escalation of the radiation treatments with low-dose 20 Gy, as well as ultra-low-dose 4 Gy radiation therapy, is under investigation within ongoing prospective clinical trials of the International Lymphoma Radiation Oncology Group (ILROG) and of the German Lymphoma Alliance (GLA).

## 1. Introduction

Marginal zone lymphoma (MZL) of mucosa-associated lymphoid tissue (MALT) belongs to low-grade B-cell lymphomas [1]. It is the most common lymphoid neoplasm arising in the mucosa and was first described in 1983 by Isaacson [2]. According to the World Health Organization (WHO), three distinct, clinically different marginal zone lymphoma (MZL) entities have been described: extranodal MZL of MALT type (MALT lymphoma), splenic MZL, and nodal MZL [3]. Extranodal marginal zone lymphomas are most frequently located in the stomach (50–86% of all cases). The most important risk factor for gastric MZL is *Helicobacter pylori* infection [3]. 

The incidence of gastric MZL (gMZL) has been increasing, and most patients present with early-stage disease. Possibly, this may be influenced by the development of advanced endoscopic ultrasound [1,4,5,6].

In *H. pylori* positive gMZL, eradication using antibiotics to remove microenvironmental stimuli supporting tumor growth results in lymphoma regression in 55.6–84.1% of cases, and a long-term complete response in approximately 75% of cases [7,8]. For patients without the successful eradication of lymphoma, or with progressive disease, treatment options have historically included surgery, whereas the current treatment modalities are immunotherapy, chemotherapy (CTx), and radiation therapy (RT) [1,9,10,11].

The optimization of the treatment strategy for gMZL has a long history. Because of the rarity of gMZL (0.4 to 0.6 cases per 100,000 persons per year) [12], there are mainly retrospective studies reporting small patient numbers. These studies combine various types of gastric non-Hodgkin lymphoma (NHL) and employ different histologic classifications, staging systems, and forms of treatment.

Prior to the early 1990s, partial or total gastrectomy was the standard of care. This procedure is associated with significant morbidity and is currently rarely used as salvage treatment [13]. Despite the lack of evidence, the main concerns about using CTx and/or RT were gastric perforation or bleeding [2,14,15,16,17,18,19]. Over time, the information improved in favor of a solely organ-preserving therapy [14,15]. Early-stage disease patients treated with RT and/or CTx showed a low incidence of severe complications and a non-inferior outcome to Sx [14,16,17,18,19,20,21,22,23,24].

The most effective single modality for local control (LC) of most types of lymphoma is radiation therapy. The history of RT in treating lymphomas shows one of the greatest successes in modern cancer treatment [25]. Because of the excellent LC compared to Sx, RT has been widely used and is internationally recommended as the therapy of choice in localized stages of lymphoma [9,26,27,28,29]. Depending on the subtype of lymphoma, the remission rates exceed 95%, but the recurrence rates increase with the length of the follow-up period. The recurrences are mainly localized or locoregional in the stomach or the duodenum [28,29,30].

From the 1960s to 1980s, the five-year overall survival rate using RT for gMZL was between 35 and 65% [15,16,31]. Currently, gMZLpatients do not usually die of their lymphoma, but reach roughly the mean life expectancy of the normal population. At 15 years post-treatment, the median age of a cohort of 178 patients (Yahalom et al.) was 78.5 years, and the life expectancy of the US population is 78.6 years [30]. 

In the past, extensive RT of the whole abdomen (WART) resulted in good local control, but also in worrisome long-term morbidity [32,33]. This prompted a renewed examination of extensive RT: reduced extended (red. EF) and involved field radiotherapy (IF), including only the initially involved regions, showed no inferior outcome to WART. The IFRT definitions were based on two-dimensional radiation therapy planned without the use of modern imaging, on bony landmarks, and on anatomical regions defined using the Ann Arbor Staging Classification system. However, although IFRT represented a significant reduction in the volume irradiated compared to the previously used EFRT, it still involved treating relatively large volumes of normal tissue, even in patients in the early stages of the disease. Today, the extensive RT fields of the past are no longer needed and the current internationally recommended treatment concept for the irradiation of gastric MZL lymphoma is an involved site radiotherapy (ISRT) with 24–30 Gy over 3 to 4 weeks [27,34,35,36] (Figure 1 and Figure 2; Table 1). Recent planning techniques attempt to further reduce the radiation dose in order to minimize the probability of normal tissue complication while maintaining tumor control [37].

This review discusses the development of indications for radiation therapy of gMZL, the dose of irradiation, the optimum treatment volume, and the related toxicity. [16]

## 2. Extended Field Radiotherapy (EFRT)

The treatment of gastric lymphoma with radiation therapy alone has been documented in the medical literature since the 1930s [59]. In 1939, Archer [59] reported on twenty gastric lymphosarcoma patients surviving 5 years after diagnosis. Eight patients were treated with biopsy and radiation alone, although this approach was commonly performed in patients with inoperable tumors.

Advances in technology enabled RT to treat large volumes, and extended field RT (EFRT), with prophylactic treatment of the entire abdomen, became the treatment of choice, thereby increasing disease-free survival (DFS) and overall survival (OS) rates [40,41,59,60,61,62,63]. Pathophysiologically, it was justified by dealing with the normal flow of the intraperitoneal fluid. Since the fluid reaches the pouch of Douglas and flows back up to the diaphragm, the gastrectomy would cause loose tumor cells to spread throughout the abdomen. The proliferation and dissemination of such cells can be prevented by whole abdominal radiotherapy (WART) [15]. Therefore, the radiation field encompassed the entire abdominal cavity in the longitudinal direction from the diaphragm to the pouch of Douglas and in transverse direction to the side wall, with dorsal shielding from the right kidney [15]. The boost covered the entire stomach, the paraaortic area to the level of L2–L3, depending on the location of the stomach, which was determined using barium meal pictures in the treatment position [21] (Figure 1a).

Most protocols used WART for primary or postoperative therapy of gastric lymphoma to a total dose of 20–30 Gy, with a sequential boost to the entire stomach bed and paraaortic node region of 40–45 Gy. Using EFRT, some studies demonstrated a survival advantage for postoperative radiation therapy [16,64,65,66,67,68,69,70,71]. Bush and Ash [66] found that WART at 25 Gy yielded a 2-year no evidence of disease survival (NED)rate of 64%, and a 2-year LC rate of 82%, compared with 44% and 36% for patients treated with resection only, respectively. Herrmann [65] applied WART at 20 Gy, followed by a boost to the stomach bed and paraaortic lymph node region, noting an 80% 5-year NED for patients solely treated with irradiation, 50% for patients solely resected, and 90% for patients with combined treatments. Similarly, Shiu [64] used WART at 25 Gy and boosted the gastric bed to 40 Gy. The five-year survival rate was 33% for patients solely resected, 67% for patients receiving postoperative irradiation, and 85% for those receiving more than 30 Gy.

In 1988, Burgers et al. [15] reported on 24 stage I gastric NHL patients who were treated with irradiation alone. The RT consisted of a three-week WART treatment at 20 Gy, followed by an additional two-week treatment at 20 Gy with a boost at 40 Gy. After a median follow-up of 48 months, the 4-year DFS was 83%. 

General prophylactic containment of the inguinal lymph nodes in the case of WART does not appear to be necessary [63].

When Fischbach [72] showed that the postoperatively irradiated patients had comparable chances of survival despite unfavorable selection criteria, such as incomplete resection, advanced stage, and other risk factors, a prospective study was carried out. 

EFRT with boosts was used in the first prospective, multicenter study, GIT NHL 01/92, initiated at the University Hospital of Muenster, Germany [14,24,65]. Whether or not the treatment included surgery was at the discretion of each participating center. After resection, patients with low-grade or indolent histological subtypes of lymphoma in stages IE and IIE were treated with WART (30 Gy) and, in case of residual disease, an additional boost with 10 Gy was used. Without gastric resection, stage IE and IIE patients received EFRT (30 Gy + 10 Gy boost) using AP/PA opposing fields with individual shielding of the kidneys and of right lobe of the liver. There were no significant differences in survival rates between patients who were resected or solely irradiated as part of their treatment. From this point on, gastrectomy was no longer integrated into the standard therapy. Currently, in gMZL, surgical management is only necessary in the case of emergency indications, such as macroscopic bleeding or perforation. 

## 3. Reduced Extended Field Radiotherapy

Shimm et al. showed that the size of the radiation fields can be reduced without affecting the prognosis in mixed gastric lymphoma. In their retrospective analysis, 19/26 patients with primary gastric lymphoma received postoperative radiation therapy. The AP-PA fields covered the gastric bed and the regional nodes (mean size, 323 cm^2^; 19 cm × 17 cm) (Figure 1). The mean dose was 36 Gy à 1.5–2.0 Gy and the 5-year OS was 58%. Three patients who received postoperative radiation therapy had abdominal failures comparable to those receiving a previous series of WART radiation therapy [71].

In accordance, Lim found that after surgery, radiation treatment at 20–30 Gy of the gastric bed and para-aortic lymph nodes improved LC from 90% to 100% in mixed gastric lymphomas [73].

In the prospective multicenter study GIT NHL 02/96 of stage I and stage II primary GI lymphomas [38,44], the aim was to de-escalate treatment. The radiation dose was 30 Gy, followed by a 10 Gy boost to the tumor region if the resection was not complete. The radiotherapy volume of patients with indolent lymphoma stage I and microscopic (R1) or macroscopic (R2) residuals after gastric resection [74] included the upper and middle part of the abdomen. The lower field boundary was the fifth lumbar vertebra (as reduced extended-field radiotherapy (red. EF), (30 Gy + 10 Gy boost to R1 or R2 regions), (Figure 1b). After complete resection, patients with stage II disease were treated with red. EF 30 Gy, while after incomplete resection (R1 or R2), the target field was a WART with 30 Gy, followed by a boost of 10 Gy to the gastric region. Non-resected patients were also treated with red. EF with 30 Gy in stage I and with WART 30 Gy in stage II. The tumor region was boosted with 10 Gy. It should be emphasized that no disadvantage could be observed with the use of an organ-preserving treatment (OS at 42 month was 86% with surgery vs. 91% without surgery; the 5-year EFS was 70%) [44]. In a prospective trial conducted by Avilés et al., 241 patients with early stage gMZL were randomized to receive surgery, radiotherapy, and chemotherapy. In the radiotherapy group, 30 Gy was administered using WART, with the liver and kidneys shielded. The upper abdomen treatment was boosted to 40 Gy. EFS after 10 years was 52% in the radiotherapy arm, 52% with surgery, and 87% in the chemotherapy group. However, the overall survival rate showed no significant differences between the three groups [46].

## 4. Involved Field Radiation Therapy (IFRT)

Maor et al. reported on a series of 34 patients with stages IE and IIE gMZL who were treated with conservative treatment alone, consisting of chemotherapy in combination with involved field radiotherapy (IFRT). The chemotherapy consisted of cyclophosphamide, doxorubicin, vincristine, prednisone and bleomycin (CHOP-Bleo); or cyclophosphamide, methotrexate, etoposide, and dexamethasone (CMED). IFRT was started after four cycles of chemotherapy; the irradiation field included the left upper quadrant (stomach, spleen, celiac, and paraaortic lymph nodes), (Figure 1c). The total dose was 30 Gy to 50 Gy at 1.8 Gy/day. A dosage exceeding 40 Gy was delivered to a reduced field that addressed the lymphoma in the stomach. Additionally, up to eight cycles of chemotherapy were administered. The 5-year OS rate was 73% and the DFS rate was 62% [39].

The successful treatment of gMZL with radiation alone at the Memorial Sloan Kettering Cancer Center was first announced in 1998 [42] and included 51 patients with *H. pylori*–independent gMZL [10,42]. The patients received 30 Gy (28.5–43.5 Gy) in 1.5-Gy doses for a period of 4 weeks to the stomach and the local lymph nodes (low dose IFRT) using opposed anterior and posterior fields. An oral (2%) barium sulfate suspension and inspiration and expiration radiographs were used to aid in localizing the stomach, and an intravenous pyelogram was used to locate the kidneys. To include the gastric lymph nodes in the treatment volume, a 2 cm margin around the gastric wall was added. 

The 5-year freedom-from-treatment failure and the overall survival rates were 89% and 83%, respectively. The cause-specific survival was 100%.

In the German multicenter prospective trial DSGL 01/2003, RT was stratified according to the stage of disease, and stage IE was treated with IFRT and IIE with red. EF. In the area of the tumor, a dose of 40 Gy was applied and 30 Gy in case of the prophylactic extended area in the red. EF, using two-dimensional (2D) opposed radiation fields or three-dimensional (3D) conformal radiotherapy (CRT) [28].

Overall, many authors have reported outstanding results after RT alone using IFRT and conventional 3D CRT [28,43,48,49,52,53,55,56,75].

## 5. Involved Site Radiotherapy (ISRT)

Modern advanced computed tomography (CT) imaging and highly conformal radiation therapy planning and delivery are currently used in patients with gMZL. Unlike most solid tumors, it is not necessary to irradiate the stomach with high doses of radiation, but rather to minimize the dose of radiation to normal tissues, as experience has shown that even relatively low doses cause significant long-term morbidity and mortality.

Current target volume and radiation dose guidelines for involved site RT (ISRT) are provided by the International Lymphoma Radiation Oncology Group (ILROG), a worldwide organization established in 2011 supporting the research on RT for lymphoma [27]. 

To date, no randomized trials comparing ISRT with IFRT have been published. It is unlikely that such studies will be conducted because, due to the low recurrence and side effect rates, a very high number of patients would have to be recruited in order to prove non-inferiority.

Rather than using the standard treatment fields of the past, ISRT is being individualized to treat each patient’s stomach and nearby lymph nodes, which may contain microscopic or macroscopic disease, in a highly conformal way using 3D imaging (Figure 1d). The ISRT concept has been accepted as the standard for modern RT for gMZL by most centers and collaborative groups, including the National Comprehensive Cancer Network (NCCN) [76].

The clinical treatment volume (CTV) for gMZL includes the stomach and first part of the duodenum. Perigastric lymph nodes and other parts of the duodenum are also included in the clinical treatment volume if they are involved by disease. Using this target volume, the irradiated volume is significantly smaller than the volume used in the old IFRT technique [27] (Figure 1a–d).

Excellent outcome has been demonstrated with ISRT using pre-defined target volume (PTV) and 30 Gy for treatment planning [30,54,55,57,77]. The highest 5-year and 10-year overall survival rates reported to date were 94% and 79%, respectively, comparable to the general population [22]. 

Intensity modulated radiation therapy (IMRT) with reduced dosage was used effectively in a recent series of 32 gMZL patients. The dose reduction to 24 Gy showed no disease failure 2 years after ISRT. The clinical target volume (CTV) for ISRT included the stomach alone for stage I or the stomach and involved lymph nodes for stage II, each with a safety margin of 2–3 cm [57].

## 6. Toxicity of Radiotherapy Treatment

The decrease in the size of the radiation fields (based on stage adaptations), along with advanced technological development, improved the ability to deliver treatment with less toxicity. Reinartz et al. assessed the toxicity of 290 patients with gMZL stage IE or IIE who were treated with radiotherapy between 1992 and 2013. Acute hemato- and gastrointestinal toxicity decreased significantly with the use of smaller radiation fields and modern radiation techniques. Chronic RT-associated side effects in organ functions were limited to a low grade and were rare [28], which is in agreement with other studies [13,30,51,53,78,79].

### 6.1. Bleeding and Perforation

For many decades, the major concern with radiation therapy and chemotherapy was the risk of fatal complications such as hemorrhage and perforation due to the malignant lesion or its therapy [80]. There are many warnings in the medical literature against treating gastric lymphoma without surgical resection, but this prevailing idea has not been confirmed in studies [68,70,81,82,83,84]. In 1990, Talamonti reported five patients with primary gastrointestinal lymphoma who initially received radiotherapy or chemotherapy and later developed severe tumor-related complications. However, all mentioned patients had advanced stage disease [85].

In contrast, in 1982, Mittal et al. showed that the frequency of perforations or bleeding due to radiotherapy is minimal. Only 1% (1/75) developed a gastric perforation directly associated with radiation therapy. Meanwhile, 10% (3/29) died of surgical complications after gastrectomy. For the first time, adjuvant radiation was recommended for gastric lymphoma in stage IE and adjuvant radiation plus chemotherapy for stage IIE [23].

Consistent with these results, Varsos and Yahalom found that the incidence of perforation in the early stage of disease when treated with radiotherapy alone is below 5% (in contrast to an operative mortality rate of 0–22%) [17], and in more recent studies, no bleeding or perforations occurred at all [28,30].

### 6.2. Renal Dysfunction

The risk of renal impairment or hypertension due to the radiation therapy of gastric lymphoma patients is low. Maor et al. examined the renal function of 27 patients with stage I or II gastric lymphoma who received at least 24 Gy on ≥1/3 of the left kidney with a median follow up at 3.4 years. Although shrinking of the ipsilateral kidney was detectable in most of the patients, only two patients developed mild hypertension. Urea or creatinine in serum was not elevated [86]. 

In WART, the right kidney should be shielded from behind [15], because if part or all of the right kidney receives a high dose of radiation, the risk of high blood pressure can increase [87].

The use of 4-field 3D CRT significantly reduces the radiation dose to the kidney. The addition of intensity modulated radiotherapy (IMRT) leads to further dose improvements for the left kidney and the liver in selected patients [45]. Using IMRT, the mean doses to the kidneys in standard dose (≥30 Gy) and reduced dose (≤24 Gy) radiotherapy can be <5 Gy, resulting in a minimal risk of renal impairment as a complication of radiotherapy [57].

In large trials, Reinartz [28] described grade 1–2 impaired chronic kidney function in only 3–7.9% of 290 gMZL patients after RT, and Yahalom [30] and Wirth [53] did not observe any late renal failure. 

### 6.3. Heart Toxicity

In the largest study on radiotherapy in gMZL, carried out by the German Study Group on Gastrointestinal Lymphoma (DSGL), 12 of 290 gMZL patients [28] treated with RT died of cardiovascular events, and in the study performed by the International Extranodal Lymphoma Study Group (IELSG), 8 of 102 gMZL patients treated with RT [53] died of cardiovascular events at a median follow-up of 6.4 years. However, in these studies, the effect of RT on cardiovascular risk remains uncertain. In Hodgkin’s lymphoma and breast cancer, RT-associated heart toxicity has long been recognized, showing a linear radiation dose-response relationship [24,25]. Given the expected long-term survival rates, minimizing radiation exposure to the heart is indispensable for reducing the risk of late cardiac events for gMZL patients. 

Due to the close proximity of the stomach to the base of the heart, motion management using the deep-inspiration breath-hold (DIBH) technique creates anatomical distance between the heart and the stomach and significantly reduces the dose of radiation to the heart [88]. Besides, modern radiation techniques and daily imaging also help to reduce the dose exposure to the heart. A Surveillance, Epidemiology, and End Results (SEER) database analysis of 2996 patients showed no increase in the risk of cardiac death among patients with stage I gMZL after radiotherapy [89].

### 6.4. Secondary Malignancy

Regardless of the type of lymphoma therapy used, the incidence of adenocarcinoma and precancerous lesions such as intestinal metaplasia (IM) after gastric lymphoma increases [29,50,90,91,92,93]. This relationship could be due to a common pathogenesis of gastric lymphoma, precancerous lesions, and adenocarcinoma as being a chronic *H. pylori* gastritis [93,94,95,96]. Another hypothesis is that the onset of IM on the gastric mucosa early after lymphoma regression could be due to destruction of the gastric glands by lymphoepithelial lesions, followed by immediate repair with intestinal cells [93]. Since the risk of gastric adenocarcinoma is described as six times higher in patients with gMZL, an accurate re-evaluation after diagnosis and treatment is warranted [5].

Although Au et al. [97] do not detect an increased incidence of secondary tumors in gMZL patients, patients with NHL are at a significantly elevated risk of secondary cancers for up to 20 years after diagnosis. The calculated risk of developing a second cancer after being diagnosed with NHL is 21% for the next 3–20 years, compared with the population-expected cumulative risk of 15% [32,33]. A larger radiation field or higher doses of radiation are important risk factors for the development of a secondary malignancy [98], and limiting these can reduce the rate of their occurrence. In agreement with this, the authors of the above-mentioned multicenter study analyzed 15 secondary malignancies discovered after WART in gMZL patients [53] and came to the conclusion that three pelvic malignancies in the entire abdomen cohort would likely have been avoided by using a limited radiation field.

### 6.5. Motion Management and Daily Imaging

Motion management using DIBH increases the distance between the base of the heart and the upper side of the stomach, leading to less radiation exposure of the heart [26,27] and limiting breath-induced gastric movement, allowing for the use of smaller PTV margins. There can be considerable interfractional fluctuations in stomach volume, even with the use of long fasting periods. Daily Image-Guided RT (IGRT) improves target coverage, despite the use of low PTV margins [57,99]. With the addition of a breath-holding technique, the PTV margins could be reduced to 0.5 to 1.0 cm for the stomach, compared to the 1.5 cm margins when the patient is breathing freely [57,79]. Retrospective analysis of daily computed tomographic (CT) scans of gastric lymphoma patients showed that a margin of 1.5–2.5 cm is required for covering 95% of the stomach due to intra- and interfractional variations of the stomach position [100,101]. The greatest deviation of the gastric position has been documented intrafractionally in the superior-inferior direction and interfractionally in the lateral direction, requiring a margin up to 3.1 cm [47].

ILROG contouring guidelines recommended that the contouring of an internal target volume be determined by 4D CT or by fluoroscopy to track the variation of stomach position [27]. When DIBH and daily volumetric imaging are not available, clinicians should consider the appropriate margins necessary to ensure consistent target coverage [27,102].

## 7. Future Directions: Standard, Intermediate, or Ultra Low-Dose Radiotherapy?

Because of the excellent outcomes of patients with gMZL after radiation treatment at 30 Gy, dose de-escalation is under consideration. A randomized trial conducted in the United Kingdom suggests that 24 Gy is effective for low-grade B-cell lymphoma. However, in this study of 248 patients who received radiotherapy, only 17% had MZL, and the frequency of gMZL remains unclear [35]. 

Pinnix et al. reported on gMZL patients treated with 24 Gy low dose ISRT using IMRT and compared them to those treated with ≥30 Gy [57]. The patients who were treated with 24 Gy (*n* = 11) showed high rates of complete response. There was no correlation between the lower dose and local recurrence at a median follow-up of 55 months. In a recent retrospective study conducted by Saifi et al. of 42 patients with early stage gMZL, reduced dose RT using 23.5–27 Gy showed comparable efficacy to standard dose RT using 30–36 Gy [58].

The HELYX II [103] trial examined the outcomes of RT in patients with persistent lymphoma after *H.pylori* eradication or in *H. pylori* negative patients. Twenty-nine low-grade gMZL lymphoma patients with stages IE and II1E lymphoma were randomized to acquire gastric RT at a dose of 25.2 or 36 Gy à 1.8 Gy. Of the 29 randomized patients, 22 patients completed the follow-up and could be analyzed after one year. No recurrences were found in either arm of the study at a median follow-up at 79 months [103].

In the International Extranodal Lymphoma Study Group (IELSG) multicenter study [53], the median total RT dose for gMZL was 40 Gy; no patient received a dose <26 Gy. In line with Pinnix [57] and Schmelz [103], no association was found between radiation field size or dosage and treatment failure.

Taking advantage of the radiosensitivity of indolent lymphoma, 4 Gy may have efficacy in many cases. It is an alternative to the current standard radiation dose of 30 Gy in cases of palliation, re-irradiation, or if the longer duration of the treatment would prevent its completion [78,104,105,106,107,108,109,110].

Haas et al. reported on 109 indolent NHL patients with 304 symptomatic sites treated with 4 Gy. The total response rate was 92%, the CR rate was 61%, and the median time to local progression was 42 months following the initial CR [110]. Other series using ultra-low dose radiotherapy in the treatment of NHL reported CR rates ranging from 37% to 84%; however, the proportion of gMZL is unknown [106,107,108,109]. 

Ultra-low dose radiotherapy is delivered over only 2 days. Patients with a poor performance status or who are traveling from long distances can easily undergo this form of treatment. However, while 4 Gy can be effective in the palliative setting, local control is significantly inferior to 24 Gy (5-year local progression-free rate is 89.9% after 24 Gy and 70.4% after 4 Gy), which remains the treatment schedule of choice for curative radiation therapy in MZL [34,74,111]. 

At present, an open-label trial of the MD Anderson Cancer Center (ClinicalTrials.gov Identifier: NCT03680586, last accessed 31 January 2022) studies how well ultra-low dose radiation with 4 Gy works in treating patients with stage I-IV gMZL. If the response to 4 Gy is inadequate, higher-dose radiotherapy may be given at the discretion of the treating physician.

However, the biological mechanism of ultra-low dose radiotherapy is not fully understood. It activates many processes that lead to cell death and apoptosis, for example, it causes the inactivation of bcl-2 overexpression [112]. As a result p53, caspase-8 and -9, might be overexpressed and macrophage activation might be upregulated [113]. The NCT03680586 study examines whether microbiome or micro-ribonucleic acid (RNA) profiles can predict the response to ultra-low dose radiation therapy. Another ongoing trial of the ILROG (ClinicalTrials.gov, Identifier: NCT04097067, last accessed 31 January 2022) assesses the correlation between blood serum biomarker levels and lymphoma response to radiation treatment.

With such potent biological effects within the cell, the rationale is that a dose greater than 4 Gy, but lower than 24 Gy, might be optimal. Hence, the open-label trial of the University Hospital of Muenster (ClinicalTrials.gov, Identifier: NCT04097067) in collaboration with the ILROG and the German Lymphoma Alliance (GLA), studies the effectiveness of intermediate low-dose radiation therapy with 10 × 2 Gy ISRT for the treatment of patients with indolent stage I–II stomach or duodenal lymphoma. 

Low-dose radiotherapy and apoptosis-inducing drugs like rituximab may be an interesting combination for gMZL treatment. In vitro experiments show that rituximab can potentiate radiation-induced apoptosis in lymphoma cells, and it acts as a radiosensitizer [114,115]. Therefore, combination regimens could result in sound local control without losing the benefits of systemic effects [116]. 

## 8. Conclusions

This review emphasizes the historically changing role of radiation therapy for gMZL, examining gastrectomy as the previously accepted treatment of choice to the current use of definitive low dose RT alone. It exemplifies the dramatic changes in radiation therapy from high-dose, large-field techniques to low-dose, localized target volumes based on advanced imaging, three-dimensional treatment planning, and advanced treatment delivery techniques, reducing toxicity while maintaining efficacy. 

This review provides compelling evidence supporting the continued use of RT as a safe and highly effective therapy for gMZL. Acute or late therapy-related toxicities are very rare, demonstrating the safety of this treatment. 

The ISRT concept for gMZL, as defined by the ILROG, individualizes the treatment of each patient’s stomach and nearby lymph nodes in a highly conformed way, using modern imaging for treatment planning and delivery. This treatment has been accepted as the standard for modern RT. With the reduction of the target volume in ISRT, the irradiation of normal tissue is significantly reduced compared to the more extensive treatment fields of the past, which leads to a reduction in the risk of long-term complications.

The evidence so far confirms these expectations, but the concept was only recently introduced. Longer periods of follow-up with careful analysis of the incidence of recurrence and the risk of long-term complications is required to assess the full effect of the ISRT concept and reduced doses in RT for gMZL.

## Figures and Tables

**Figure 1 cancers-14-00873-f001:**
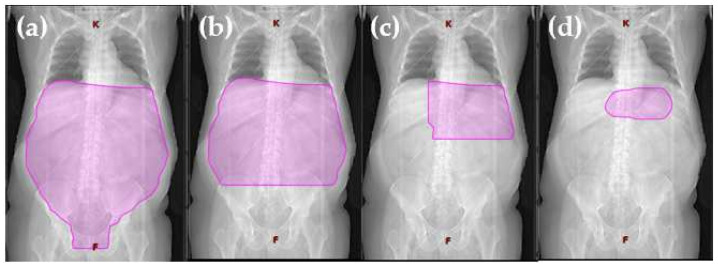
Visualization of radiation volume decrease from extended field to involved site radiotherapy. Definition of field sizes (**a**) Extended field (Burgers et al., 1988 [15]): the entire peritoneal cavity from the diaphragm to the pouch of Douglas and laterally to the side wall. (**b**) Reduced extended field (Willich et al., 2000 [38]): in the case of complete resection of a gastric tumor smaller than 5 cm in diameter, without submucosal perforation, the target volume was restricted to the upper and middle part of the abdomen, sparing the pelvis. (**c**) Involved field (Maor et al., 1990 [39]): the left upper quadrant (stomach, spleen, celiac, and paraaortic lymph nodes). (**d**) Involved site radiotherapy (ILROG guidelines Yahalom et al., 2015 [27]): the location is individualized to treat each patient’s stomach and nearby lymph nodes, which can contain microscopic or macroscopic disease, in a highly conformal way using 3D imaging.

**Figure 2 cancers-14-00873-f002:**
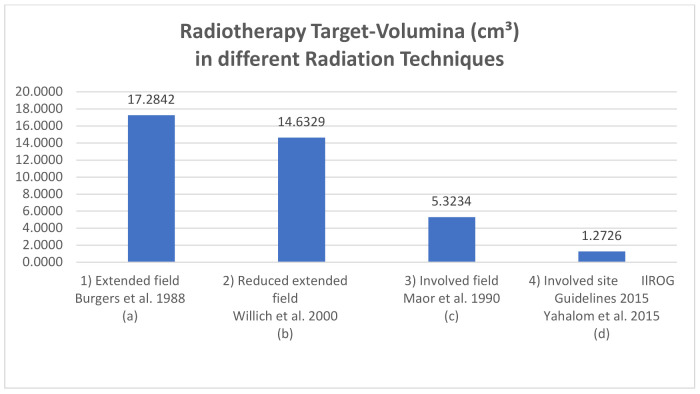
Exemplary RT-volume (cm^3^) in different radiation techniques measured with a 3D-radiation planning program. Definition of field sizes (**a**) Extended field (Burgers et al., 1988 [15]): the entire peritoneal cavity from the diaphragm to the pouch of Douglas and laterally to the side wall. (**b**) Reduced extended field (Willich et al., 2000 [38]): in the case of complete resection of a gastric tumor smaller than 5 cm in diameter, without submucosal perforation, the target volume was restricted to the upper and middle part of the abdomen, sparing the pelvis. (**c**) Involved field (Maor et al., 1990 [39]): the left upper quadrant (stomach, spleen, celiac, and paraaortic lymph nodes). (**d**) Involved site radiotherapy (ILROG guidelines Yahalom et al., 2015 [27]): the location is individualized to treat each patient’s stomach and nearby lymph nodes, which can contain microscopic or macroscopic disease, in a highly conformal way using 3D imaging.

**Table 1 cancers-14-00873-t001:** The development of total radiation doses for the treatment of gMZL from the 1990s to the present day.

Publication Author	Publication Year	References	Study Nature	High Grade NHL Included in N	*N*	Radiation Dose (Gy)	Single Dose (Gy)
Taal	1993	[40]	Retrospective	No	42	40	2.0
Kocher	1997	[41]	Retrospective	No	25	30–40	1.5–2.0
Schechter	1998	[42]	Retrospective	No	17	median 30 (28.5–43.5)	1.5
Tsang	2001	[43]	Retrospective	No	9	median 25 (20–30)	1.0–2.5
Koch	2001	[14]	Prospective	Yes	106	40	1.5–2.0
Koch	2005	[44]	Prospective	No	143	40	1.5–2.0
Della Biancia	2005	[45]	Retrospective	No	14	30	Not available
Avilés	2005	[46]	Prospective	No	78	40	Not available
Watanabe	2008	[47]	Retrospective	No	11	30	1.5
Vrieling	2008	[48]	Retrospective	No	115	40	1.0–2.0
Tomita	2009	[49]	Retrospective	No	20	median 32 (25.6–50)	1.5–2.2
Ono	2010	[50]	Retrospective	No	8	30	1.5
Zullo	2010	[7]	Retrospective	No	112	median 30 (22.5–43.5)	1.5–1.8
Goda	2010	[51]	Retrospective	No	25	median 30 (17.5–35)	2.5
Fischbach	2011	[52]	Prospective	No	19	46	1.8–2.0
Wirth	2013	[53]	Retrospective	No	102	median 40 (26–46)	median 1.8
Abe	2013	[54]	Retrospective	No	34	30	1.5–2.0
Teckie	2015	[29]	Retrospective	No	123	median 30 (Range unknown)	2.0
Ruskone-Fourmestraux	2015	[55]	Prospective	No	232	30	2.0
Ohkubo	2017	[56]	Retrospective	No	27	median 30 (30–39.5)	1.5
Pinnix	2019	[57]	Retrospective	No	32	median 30 (24–36)	1.5
Reinartz	2019	[28]	Prospective	No	290	median 40 (36–44)	1.8–2.0
Yahalom	2021	[30]	Retrospective	No	178	median 30 (22.5–43.5)	1.5
Saifi	2021	[58]	Retrospective	No	42	median 30 (23.5–36)	1.5–2.0

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
