# Peer review of "Development of Organ-Preserving Radiation Therapy in Gastric Marginal Zone Lymphoma"

_cancers, 2022, doi:10.3390/cancers14040873_

Round 1

Reviewer 1 Report

In this manuscript Rolf et al. review advances in radiation therapy in patients with gastric EMZL. The manuscript is well written and important for the lymphoma community.

I have the following comments

First paragraph in page #2 authors include surgery as a treatment option in gastric EMZL. I suggest to remove surgery as treatment option in this disease.

in table 1 please include the nature of these studies (retrospective/prospective)

first paragraph in page 6, authors describe OS at 42 month of 86% with surgery vs 91% without surgery. In EMZL will be more relevant to describe the PFS or EFS rather than OS. 

In toxicity section of radiotherapy, will be relevant to include the median follow up of these studies as many events may not be censored due to short follow ups

In heart toxicity AE, the authors describe two studies with 4% and almost 8% of enrolled patients dying from heart toxicity. this is a relevant toxicity in an indolent disease such as EMZL. please further describe the reasons for this toxicity and modern approaches to avoid these toxicities.

please correct typos present in page #9 (Future directions)

please provide references of studies demonstrating radiosensitizer effect of rituximab 

will be interesting if authors can expand in how new technologies/approaches can be integrated in the management of patients with gastric EMZL including proton therapy and others. 

Author Response

Reviewer 1:

In this manuscript Rolf et al. review advances in radiation therapy in patients with gastric EMZL. The manuscript is well written and important for the lymphoma community.

First paragraph in page #2 authors include surgery as a treatment option in gastric EMZL. I suggest to remove surgery as treatment option in this disease.

Removed.

In table 1 please include the nature of these studies (retrospective/prospective)

Included.

first paragraph in page 6, authors describe OS at 42 months of 86% with surgery vs 91% without surgery. In EMZL will be more relevant to describe the PFS or EFS rather than OS. 

I agree that PFS and EFS are relevant factors. According to Koch et al. we added the EFS with surgery vs. without surgery.

In toxicity section of radiotherapy, will be relevant to include the median follow up of these studies as many events may not be censored due to short follow ups

The specified median follow ups have been added.

In heart toxicity AE, the authors describe two studies with 4% and almost 8% of enrolled patients dying from heart toxicity. this is a relevant toxicity in an indolent disease such as EMZL. please further describe the reasons for this toxicity and modern approaches to avoid these toxicities.

Thank you for your advice. A causal relationship with RT is uncertain. The SEER database results showed no increase in cardiac death among gMZL patients after RT. We provided more detailed information.

please correct typos present in page #9 (Future directions)

Corrected.

please provide references of studies demonstrating radiosensitizer effect of rituximab 

We provided two references.

will be interesting if authors can expand in how new technologies/approaches can be integrated in the management of patients with gastric EMZL including proton therapy and others. 

We provided more information about new technologies, in particular motion management, daily imaging and low dose radiotherapy. All studies on gastric MALT lymphoma applied photon-therapy (Table 1). There is limited data on proton beam therapy in the gastric region. This may be due to the high inter- and intrafractional movement of the gaster, which can lead to under- or over dosages of the planning target volume due to long treatment time with proton therapy. While breathing, the heart could also move into the sharply defined proton radiation field, potentially increasing the risk of cardiovascular morbidity.

Reviewer 2 Report

This is an excellent review of the progressive de-escalation of therapy for patients with gastric MALT lymphoma. The authors have carefully collected relevant references and described the step-wise improvements in care for these patients.  This work will be an important addition to the literature.

General comments:

It might be useful to have someone help with copy editing.  There are some wording choices that could be improved and some punctuation additions that would make this easier to read.  I will put examples below.

I think the abstract and the introduction could benefit from slight re-organization.  I would first describe the change from surgical management to RT-based (+-chemo) therapy.  Then describe the progressive RT advances.  It jumps around a bit. 

Suggested changes by section:

Abstract:

"RT accompanied by fear of perforation and bleeding omitting surgery"

24-30 gy, see below

Intro:

For sentence"regardless of stage" I would clarify that this is for Hpylori + cases.  What is clinical control?

"since eradication of lymphoma...."  could change to For patients without successful eradication of lymphoma or progressive disease...treatment options have historically included surgery..."

Change "conservative therapy" to organ-preserving?

Change "latest" to modern

Would add "overall" survival to "while in the 1960....the 5 year survival"

The paragraph that starts with this sentence is confusing and could use clarification

The paper references 30 gy as standard, but 24 Gy is now acceptable per NCCN.  Please add this. 

For the sentence "recent approaches try to further..." I would clarify that you are talking about planning techniques. 

Figure 1.  I would add to point 4: "whole stomach and nearby involved lymph nodes"

The caption for Figure 2 is repetitive with Fig 1 and can be removed/ modified.

For Table 1, some of these studies had dose ranges.  The median and dose range of all studies should be reported. 

“Pathophysiological” should have “ly”

In general, I would clarify for each study whether patients with high grade NHL were included.  This has drastic implications for outcomes in studies that were not incorporating chemotherapy.

The comment “reserved for macroscopic bleeding or perforation...”  I think I would qualify this.  Patients might require surgical management, but still not likely total gastrectomy.

Reduced ex field radiotherapy

Again, please clarify if high grade, low grade or mixed

“macroscopic (R2) rest…”  I think rest needs to be changed.

  1. ISRT

“Excellent outcomes with ISRT have been demonstrated”

Should “largest 5-year” be highest?

“ISRT includes in the clinical target volume….” I would changed this to included and clarify safety margin

References 70 and 72 are the same

Another recent publication on the topic: PMID: 34409210

  1. Toxicity of RT treatment

The major concern against – change to with

Please clarify “but the prevailing impression is not well document”

Paragraph beginning with “consistent with it”:  did the perforations lead to death?  If not please clarify as much different that operative mortality

Any role for discussing the use of endoscopic US in gastric MALT?  Some routinely employ it.

6.3 Heart toxicity

I think something is missing after (DSGL) 12 of 290 …

6.4 Secondary malignancy

I would change arises to increases

Any role for discussing intestina metaplasia commonly seen in gastric MALT pts?

Change “as may be”

6.5 Motion management

Need to add discussion of specifically CT based IGRT and the necessity for accurate alignment.

Additional relevant references for consideration:

Johnson ME, Pereira GC, El Naqa IM, Goddu SM, Al-Lozi R, Apte A, Mansur DB. Determination of planning target volume for whole stomach irradiation using daily megavoltage computed tomographic images. Pract Radiat Oncol. 2012 Oct-Dec;2(4):e85-e88.

Watanabe M, Isobe K, Uno T, Harada R, Kobayashi H, Ueno N, Ito H. Intrafractional gastric motion and interfractional stomach deformity using CT images. J Radiat Res. 2011;52(5):660-5. Epub 2011 Sep 1.

Reinartz G, Haverkamp U, Wullenkord R, Lehrich P, Kriz J, Buther F, Schafers K, Schafers, Eich HT, 4D-Listmode-PET-CT and 4D-CT for optimizing PTV margins in gastric lymphoma Determination of intra- and interfractional gastric motion. Strahlenther Onkol (2016) 192:322–332.

Wang H, Milgrom SA, Dabaja BS, Smith GL, Martel M, Pinnix CC. Daily CT guidance improves target coverage during definitive radiation therapy for gastric MALT lymphoma. Pract Radiat Oncol. 2017 Nov - Dec;7(6):e471-e478. PMID: 28377138

Hu W1, Ye J, Wang J, Xu Q, Zhang Z. Incorporating breath holding and image guidance in the adjuvant gastric cancer radiotherapy: a dosimetric study. Radiat Oncol 2012 Jun 20;7:98

  1. Future directions

Would add the updated outcomes from the FORT trial

Would discuss the adapted approach used in some of the attempts to use 4 gy, giving additional dose if inadequate response to 4 gy as in referenced MDACC trial

Conclusion

Would remove “this is the most detailed review”

Dramatically should be dramatic

Author Response

This is an excellent review of the progressive de-escalation of therapy for patients with gastric MALT lymphoma. The authors have carefully collected relevant references and described the step-wise improvements in care for these patients.  This work will be an important addition to the literature.

General comments:

It might be useful to have someone help with copy editing.  There are some wording choices that could be improved and some punctuation additions that would make this easier to read.  I will put examples below.

Language has been edited.

I think the abstract and the introduction could benefit from slight re-organization.  I would first describe the change from surgical management to RT-based (+-chemo) therapy.  Then describe the progressive RT advances.  It jumps around a bit. 

We slightly re-organized the abstract and the introduction.

Suggested changes by section:

Abstract:

"RT accompanied by fear of perforation and bleeding omitting surgery"

We have removed this part of the sentence.

24-30 gy, see below

We changed the dose to 24-30 Gy.

Intro:

For sentence"regardless of stage" I would clarify that this is for Hpylori + cases.  What is clinical control?

We clarified that this is for H. pylori + patients and changed clinical control to complete response.

"since eradication of lymphoma...."  could change to For patients without successful eradication of lymphoma or progressive disease...treatment options have historically included surgery..."

We revised this sentence.

Change "conservative therapy" to organ-preserving?

We changed “conservative therapy" to organ-preserving.

Change "latest" to modern

Changed.

Would add "overall" survival to "while in the 1960....the 5 year survival"

We added “overall”.

The paragraph that starts with this sentence is confusing and could use clarification

We clarified and re-wrote this paragraph.

The paper references 30 gy as standard, but 24 Gy is now acceptable per NCCN.  Please add this. 

We added this and referred to NCCN Guidelines.

For the sentence "recent approaches try to further..." I would clarify that you are talking about planning techniques. 

We clarified this sentence.

Figure 1.  I would add to point 4: "whole stomach and nearby involved lymph nodes"

We added involved.

The caption for Figure 2 is repetitive with Fig 1 and can be removed/ modified.

We removed the repetitive caption.

For Table 1, some of these studies had dose ranges.  The median and dose range of all studies should be reported. 

We updated the table and reported all available medians and dose ranges.

“Pathophysiological” should have “ly”

Edited.

In general, I would clarify for each study whether patients with high grade NHL were included.  This has drastic implications for outcomes in studies that were not incorporating chemotherapy.

 We added "high grade NHL were included: yes/no" to table 1.

The comment “reserved for macroscopic bleeding or perforation...”  I think I would qualify this.  Patients might require surgical management, but still not likely total gastrectomy.

Nowadays, in gMZL surgical mangagment is necessary in emergency indications such as macroscopic bleeding or perforation.

Reduced ex field radiotherapy

Again, please clarify if high grade, low grade or mixed

We clarified this.

“macroscopic (R2) rest…”  I think rest needs to be changed.

 We changes this to residuals.

  1. ISRT

“Excellent outcomes with ISRT have been demonstrated”

We changes this.

Should “largest 5-year” be highest?

Yes, we corrected this.

“ISRT includes in the clinical target volume….” I would changed this to included and clarify safety margin

We changed this and clarified the safety margin.

References 70 and 72 are the same.

We corrected this.

Another recent publication on the topic: PMID: 34409210

Thank you! We included this important study to our review and table 1.

  1. Toxicity of RT treatment

The major concern against – change to with

Changed.

Please clarify “but the prevailing impression is not well document”

We clarified the sentence.

Paragraph beginning with “consistent with it”:  did the perforations lead to death?  If not please clarify as much different that operative mortality

We clarified this.

Any role for discussing the use of ?  Some routinely employ it.

          This review emphasizes the historically changes and recent approches of radiotherapy in gMZL. Detailed discussion of Epidemiology, Pathology, Staging, Antibiotic therapy and follow up is provided by ESMO Consensus conferences: guidelines on malignant lymphoma[1]. This reference has been added in the introduction after Possibly, this may be influenced by advanced endoscopic ultrasound.”

6.3 Heart toxicity

I think something is missing after (DSGL) 12 of 290 …

We edited this.

6.4 Secondary malignancy

I would change arises to increases

We changed this.

Any role for discussing intestina metaplasia commonly seen in gastric MALT pts?

This is an important point. We added this to the part “secondary malignancies”

6.5 Motion management

Need to add discussion of specifically CT based IGRT and the necessity for accurate alignment.

Additional relevant references for consideration:

Johnson ME, Pereira GC, El Naqa IM, Goddu SM, Al-Lozi R, Apte A, Mansur DB. Determination of planning target volume for whole stomach irradiation using daily megavoltage computed tomographic images. Pract Radiat Oncol. 2012 Oct-Dec;2(4):e85-e88.

Watanabe M, Isobe K, Uno T, Harada R, Kobayashi H, Ueno N, Ito H. Intrafractional gastric motion and interfractional stomach deformity using CT images. J Radiat Res. 2011;52(5):660-5. Epub 2011 Sep 1.

Reinartz G, Haverkamp U, Wullenkord R, Lehrich P, Kriz J, Buther F, Schafers K, Schafers, Eich HT, 4D-Listmode-PET-CT and 4D-CT for optimizing PTV margins in gastric lymphoma Determination of intra- and interfractional gastric motion. Strahlenther Onkol (2016) 192:322–332.

Wang H, Milgrom SA, Dabaja BS, Smith GL, Martel M, Pinnix CC. Daily CT guidance improves target coverage during definitive radiation therapy for gastric MALT lymphoma. Pract Radiat Oncol. 2017 Nov - Dec;7(6):e471-e478. PMID: 28377138

Hu W1, Ye J, Wang J, Xu Q, Zhang Z. Incorporating breath holding and image guidance in the adjuvant gastric cancer radiotherapy: a dosimetric study. Radiat Oncol 2012 Jun 20;7:98

We have written the paragraph in more detail and added all additional references. 

  1. Future directions

Would add the updated outcomes from the FORT trial

We added the updated outcomes from the FORT trial.

Would discuss the adapted approach used in some of the attempts to use 4 gy, giving additional dose if inadequate response to 4 gy as in referenced MDACC trial

We added this important point to the section "future directions".

Conclusion

Would remove “this is the most detailed review”

Removed.

Dramatically should be dramatic

Done.

References

1.            Dreyling, M.; Thieblemont, C.; Gallamini, A.; Arcaini, L.; Campo, E.; Hermine, O.; Kluin-Nelemans, J.C.; Ladetto, M.; Le Gouill, S.; Iannitto, E.; et al. ESMO Consensus conferences: guidelines on malignant lymphoma. part 2: marginal zone lymphoma, mantle cell lymphoma, peripheral T-cell lymphoma. Ann. Oncol. 2013, 24, 857–877, doi:10.1093/annonc/mds643.